# Generative Trace Attribution Network

## Abstract

A deepfake is a digitally created or altered image, video, or audio made using artificial intelligence (AI) that seems real but is intended to deceive or mislead viewers. The rapid rise of generative AI tools like GANs and diffusion models has made it easier to create deepfake images. While traditional methods can detect deepfakes, they often fail to identify which model created them. The process of matching a deepfake image to its generative model is known as deepfake attribution. Deepfake attribution is essential for accountability and preventing misuse of generative models in the creation of deefake. Unfortunately, most existing attribution methods only work well on the specific models they were trained on. They struggle to attribute images generated from unseen generators with different initialization seeds, trained for additional epochs, fine-tuned, retrained, or having slight modifications in loss functions or model architecture. To address these limitations, we propose the Generative Trace Attribution Network (GTA-Net), a generalized attribution network that robustly attributes fake images across diverse generative models with variations, including entirely unseen generative models. GTA-Net works by analyzing hidden patterns in input images using a combination of frequency analysis and latent space analysis to capture training-induced artifacts of target generative models to be attributed. GTA-Net also employs supervised contrastive learning to separate features between different target generative models. Extensive experiments on diverse generative models demonstrate that GTA-Net significantly outperforms existing attribution techniques, offering a more robust and reliable approach for deepfake attribution.

## 1 Introduction

Deepfake technology, powered by generative models such as Generative Adversarial Networks ($GAN$s) and diffusion models, enables the creation of hyper-realistic digital media that is nearly indistinguishable from real content Karras et al. (2021b); Ho et al. (2020a). While this technology has beneficial applications in entertainment, virtual reality, and education, it poses serious security risks. Deepfakes are increasingly exploited for misinformation campaigns France24, defamation NPR, identity theft CBS News, harassment NBC News, threatening public trust and individual privacy Amezaga & Hajek (2022); Chauhan (2023). Deepfake techniques can generally be classified into two categories: manipulation-based and synthesis-based. In manipulation-based deepfake techniques, the original image is altered using deep learning techniques such as face-swapping Perov et al. (2020), face reenactment Thies et al. (2016), and attribute manipulation Choi et al. (2020). Since these modifications rely on real images, forensic analysis can sometimes identify inconsistencies, such as unnatural facial expressions, blending artifacts, or lighting mismatches Guarnera et al. (2020); Chai et al. (2020); Wang et al. (2020). In contrast, synthesis-based techniques generate entirely new images from scratch Karras et al. (2021b; 2017); Ho et al. (2020a) without requiring any real image, making detection significantly more challenging.

Deepfake detection is generally framed as a binary classification problem, distinguishing between real and fake images. Classification alone is inadequate, as it fails to provide accountable evidence of tampering or image synthesis to the forensic team. Therefore, manipulation-based deepfake detection employs localization techniques Chen & Others (2022); Mazaheri & Roy-Chowdhury (2022) to identify tampered regions by analyzing inconsistencies in the image. However, synthesis-based deepfakes generate images entirely from

scratch, making localization techniques ineffective. Thus, forensic team use model attribution to analyze subtle artifacts, statistical patterns, and architectural fingerprints embedded in generated images to infer the identity of the generative model used Kim et al. (2020); Marra et al. (2019). This process is analogous to ballistic forensics, where microscopic markings left on a bullet are examined to trace it back to the specific firearm that fired it. Moreover, training high-quality generative models requires extensive computational resources, large datasets, and expert knowledge, limiting the availability of open-source models. Thus, many state-of-the-art models remain proprietary and is a valuable assets. Therefore, there must be a copyright protection mechanism for these generative models. In this context, model attribution can be used to validate the ownership of a generative model by extracting and analyzing the model fingerprints from its generated image.

Traditional attribution techniques rely on two aspects for identifying the generated images where the first one involves embedding a unique fingerprint in the training data such that the generated image will contain the same fingerprint Kim et al. (2020); Yu et al. (2020; 2021), while the second approach focuses on identifying the unique patterns left behind by different generative models on generated images Marra et al. (2019); Xuan et al. (2019); Joslin & Hao (2020); Frank et al. (2020); Yu et al. (2019); Ding et al. (2021); Jeon et al. (2020); Li et al. (2024); Sun et al. (2023); Girish et al. (2021); Yang et al. (2023); Zheng et al. (2025). However, these techniques often lack robustness and fail to generalize across models as they are trained on the images from a specific instance of the generative model. One can bypass this technique by fine-tuning the generative model, retraining it on a different dataset, or tweaking hyperparameters, which can create an infinite number of unseen variations, making attribution inconsistent Yu et al. (2019). This highlights the limitations of existing methods, as visually demonstrated in Figure 1. These challenges point to the need for a more generalized and robust attribution technique that remains effective.

We propose $GTA\text{-}Net$ (Generative Trace Attribution Network), a robust deepfake attribution technique that generalizes across retrained, fine-tuned, modified or unseen generative models. Unlike prior methods that identify specific model instances, $GTA\text{-}Net$ attributes images by learning discriminative representations that remain informative across a range of model variations. $GTA\text{-}Net$ has three key components: an Abstractor Network, a Feature Extraction Network, and a Multi-class Classification Network. The Abstractor Network first processes each input image using a denoising autoencoder and wavelet transform to extract residual and frequency-based features that capture subtle traces left by the generative model, while minimizing the influence of image content. These extracted residual and frequency-based features are combined into a unified representation and then refined using a multi-head self-attention mechanism, which helps the network focus on the most informative patterns corresponding to generative models to be attributed. Next, the Feature Extraction Network uses contrastive learning, a self-supervised learning technique, given a pair of images whose objective is to push the input images closer or further apart depending on whether the images are from the same or different generative models. This helps the network create precise and meaningful feature representations that capture the unique characteristics of each generative model. Finally, the Classification Network uses these features to accurately identify which generative model produced the image. Our experimental results demonstrate that the sequential architecture of $GTA\text{-}Net$ facilitates the extraction of model-specific patterns, enabling robust attribution even in challenging scenarios, such as changes in seeds, datasets, training duration, loss functions, minor architectural variations or encountering unknown generative models. In summary, we make the following contributions:

- We proposed $GTA\text{-}Net$, which aims to attribute fake images to their source generative model, irrespective of whether the models producing those images have been retrained with an alternative seed, fine-tuned for additional epochs, trained on different datasets, use slightly different loss functions or architecture.

- $GTA\text{-}Net$ facilitates attribution of unseen generative models (different generative models whose generated data is not used in training) by identifying the closest matching generative model using supervised contrastive learning, frequency-based analysis, and latent space analysis.

- We conducted comprehensive evaluations of $GTA\text{-}Net$ on various generative models to demonstrate the accuracy and robustness of our technique.

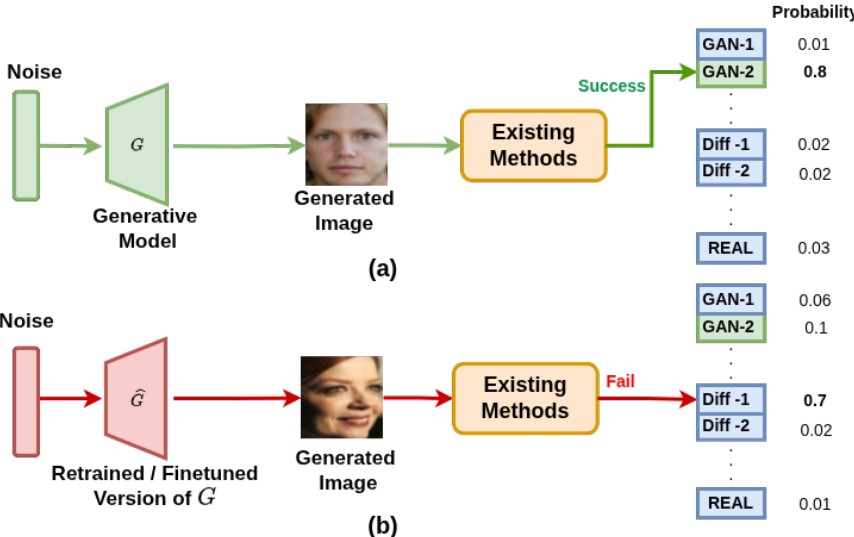

Figure 1: Key drawbacks of existing methods. In (a), attribution is correctly performed on the original instance of the generative model. However, as shown in (b), when the generative model is fine-tuned or retrained, the existing technique fails to attribute correctly. The green box is the ground truth label.

## 2 Related Work

Various deepfake detection techniques have been developed to differentiate real images from fake images Guarnera et al. (2020); Zhao et al. (2021); Asnani et al. (2023); Jeon et al. (2020); Durall et al. (2020); Wang et al. (2020). Simply identifying a generated image as fake is not sufficient, as it does not provide any insights into its origin. Forensic reliability is significantly improved when attribution techniques not only detect a fake but also attribute it back to the generative model. This deeper attribution helps distinguish between different generative models, enabling better accountability, forensic analysis, and mitigation of misuse. The attribution of generated images falls into two main categories: artificial fingerprint embedding and model fingerprinting. The first category of artificial fingerprint embeds unique fingerprints into the training data so that any generated image retains identifiable fingerprints Kim et al. (2020); Yu et al. (2020; 2021). These fingerprints can then be extracted from generated images to attribute them to a specific generative model. However, this technique requires white-box access to the model, which limits its applicability in real-world scenarios. Additionally, it does not work for pre-trained models that were not trained with artificial fingerprint-embedded datasets, and retraining existing models to incorporate fingerprints is computationally expensive and time-consuming. In contrast, model fingerprinting identifies unique patterns and artifacts naturally left by different generative models in their generated images without altering the training process. These patterns might arise from architectural differences and training procedures, which remain consistent even when models are trained on different datasets. This technique operates in a black-box setting, making it more suitable for real-world applications.

The model fingerprinting is performed using both statistical and deep learning-based techniques. In statistical techniques, the focus is on detecting residual noise in generated images using various filters Marra et al. (2019), handcrafted features Li et al. (2024) or performing frequency analysis through techniques like the Discrete Cosine Transform ($DCT$) Frank et al. (2020) and Discrete Fourier Transform ($DFT$) Joslin & Hao (2020). Deep learning-based techniques, on the other hand, utilize image transformation techniques, filters, residual images, and custom loss functions to capture subtle, architectural and training-specific features. These features are then passed into a classifier for final attribution Xuan et al. (2019); Yu et al. (2019); Yang et al. (2022). However, existing techniques perform attribution on seen models, meaning both the training and testing data are generated from the same trained generative model. These techniques fail when tested on a retrained or fine-tuned version of the same generative model. Few techniques Yang et al. (2022) attempts to improve robustness via patch-wise contrastive learning. While effective in some cases, it is sensitive to

resolution changes, making it less suitable for diffusion models that produce high-resolution, low-artifact images. Recent works Sun et al. (2023); Girish et al. (2021); Yang et al. (2023); Zheng et al. (2025) explore open-world attribution, where test models may be unseen during training. While promising, these methods often depend on distributional assumptions or thresholds that struggle with fine-tuned variants and fail to generalize to realistic, artifact-free images from advanced generative models. A technique Mu et al. (2025) also tried to attribute AI-generated ex-regulatory images by extracting model fingerprint features using the CLIP network, but it limits the attribution coverage of unseen real-world distributions. Recent attribution frameworks such as MAID Zhu et al. (2025) and FRIDA Bonechi et al. (2025) leverage diffusion-model representations to identify source generators. However, their feature extraction mechanisms are inherently diffusion-centric and have been primarily evaluated on GAN and diffusion-based generators. Consequently, their ability to generalize to emerging generative paradigms, including autoregressive and flow-matching models, remains unclear.

To address the limitations of both patch dependent and open-world attribution methods, we propose $GTA\text{-}Net$, a robust and resolution-independent model attribution framework. Unlike $DNA\text{-}DET$, which relies on patch-wise contrastive learning and is sensitive to changes in image resolution and structure, $GTA\text{-}Net$ operates holistically without decomposing images into patches. It begins by extracting content-dependent features, allowing the network to isolate and amplify generative model-specific traces that persist across different datasets, fine-tuning, and retraining. This makes $GTA\text{-}Net$ inherently more robust to model modifications and domain shifts. Crucially, $GTA\text{-}Net$ is designed to capture features that are less sensitive to changes in training conditions and moderate architectural variations, thereby supporting robust attribution across both seen and unseen generative models. This contrasts with existing open-world attribution methods, which often depend on handcrafted thresholds or distributional assumptions, making them vulnerable to minor perturbations or high-fidelity outputs from modern diffusion models. By learning discriminative features, $GTA\text{-}Net$ achieves superior performance in both closed-world and open-world settings, offering reliable attribution even under black-box constraints and with photorealistic inputs.

## 3 Proposed Methodology

### 3.1 Problem Formulation

The growing popularity of generative models, particularly those capable of creating realistic images from scratch, has sparked significant interest in their applications across various domains. These include the generation of avatars, virtual environments, $AI$-assisted content creation, synthetic medical scans, artwork, and more. As a result, a wide range of generative models with high-quality image generation capabilities has been developed. However, with the growing prevalence of generative models, attributing generated images to their source is essential for ensuring reliability, preventing misuse, maintaining accountability, and fostering transparency in $AI$-generated content. To address this, we define attribution as the task of determining which source generative model produced a given fake image. This can be framed as a multi-class classification problem, where the goal is to attribute each image to its generative model or identify it as real. Given an image $X^y$ with source $y \in \{real, G_1, G_2, \ldots, G_N\}$, where $\{G_1, G_2, ..., G_N\}$, are different generative models. Our goal is to learn a mapping $f(X^y) \rightarrow y$.

### 3.2 Key Challenges

Formulating a multiclass classification problem for generative model attribution may appear straightforward. However, the slightest change, such as fine-tuning, training for additional epochs or retraining, will lead to different instances of the original generative model, which will fail the multiclass classifier. One can even think of defining distinct classes for each of those model instances. In that case, the key challenge is to determine what constitutes a distinct generative model. For instance, does modifying architectural components, such as adding or removing layers or altering the loss function, fundamentally constitute a distinct model? Furthermore, variations introduced through different initial seeds, datasets, training epochs, optimization strategies, or fine-tuning can lead to substantially different model instances, raising the question of how many such variations are necessary to construct a robust classifier that generalizes across all possible

instances of the target generative models. Adding to that, in real-world scenarios, we are more likely to encounter previously unseen models or variants. Therefore, the core challenge is to distinguish among known variants but also attribute outputs to the correct generative models based on underlying architecture and training pipelines.

To address these challenges, $GTA\text{-}Net$ is designed to attribute images to the closest matching generative model rather than assigning them to a generic unknown class, enabling a more precise inference of the underlying architectural family and training paradigm. For previously unseen models, $GTA\text{-}Net$ attributes images to the closest learned representation in the embedding space, which often corresponds to models with similar generative characteristics. This approach preserves structural similarity information, improves interpretability, and allows finer-grained forensic analysis even when the exact generator was unseen during training. To enhance generalization and mitigate model-specific biases, multiple instances of each generative model are trained while varying key hyperparameters such as learning rate, batch size, and training steps. This encourages $GTA\text{-}Net$ to learn representations that generalize beyond specific generative model instances.

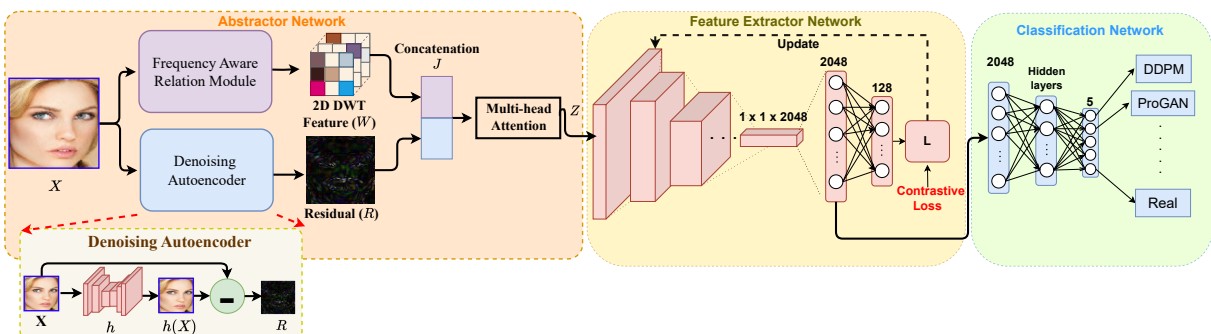

Figure 2: GTA-Net architecture for generative model attribution. It consists of three modules: Abstractor Network for extracting residual and frequency features using a denoising autoencoder and wavelet transform; Feature Extraction Network, which refines representations via contrastive learning; Multi-Class Classification Network, to perform final attribution for identifying the source generative model.

### 3.3 Overview

The technique of $GTA\text{-}Net$ can be seen in Fig. 2. $GTA\text{-}Net$ contains three sub-networks: the first one is an abstractor network, the second one is a feature extraction network, and the third one is a multi-class classification network. The abstractor network, feature extraction network, and multi-class classification network are trained as a pipeline. First, the abstractor network is trained on the fake images from different generative models and real images such that it can extract data-independent traces from images, which focus more on the discriminative representations that remain informative across a range of model variations rather than the type of data generated by the model. The feature extraction network is trained on these traces to learn feature embeddings that encode attribution-relevant characteristics of different generative models, facilitating robust discrimination between model families and variants. The multi-class classification network is trained using the features obtained from the feature extraction network for final attribution. The details of each network are discussed in the following section.

### 3.4 Abstractor Network

The objective of the abstractor network is to reduce data dependency and focus more on the discriminative representations that remain informative across a range of generative models. It consists of the following two modules:

### 3.4.1 Denoising Autoencoder

The aim of this module is to extract high-level features that are independent of image content and can serve as a fingerprint for attributing generative models. Our use of reconstruction residuals is inspired by prior forensic studies demonstrating that generator-specific artifacts are more visible in residual or frequency representations than in RGB space. Marra et al. (2019) and Frank et al. (2020) showed that image residuals and spectral artifacts provide discriminative cues for source attribution. More recently, MAID Zhu et al. (2025) demonstrated that diffusion-model reconstruction errors and inverse-diffusion activations contain generator-specific discriminative representations useful for attribution. Motivated by these findings, we employ a denoising autoencoder to isolate reconstruction residuals that emphasize generator traces while suppressing semantic content. To extract high-level features, we first train a denoising autoencoder on real images. Once trained, the autoencoder takes an input image, whether real or generated and reconstructs it. Next, we compute the residual image by subtracting the reconstructed image from the original input image. These residuals capture subtle differences that are partly dependent on the image content but can also contain unique traces left by the generative model. The task of the denoising autoencoder can be mathematically formulated as:

$$R = abs(X - h(X)) \tag{1}$$

where $X$ is the input image, $h$ represents the trained denoising autoencoder which reconstructs $X$, $R$ is the calculated residual image, and $abs$ represents absolute value. From a representation-learning perspective, the denoising autoencoder approximates a projection onto the manifold of natural images learned from real data. Let $(\mathcal{M})$ denote this manifold and $(h(X))$ the reconstruction produced by the denoising autoencoder. The residual R therefore measures the component of the image that lies outside $(\mathcal{M})$. Since generative models introduce characteristic synthesis artifacts arising from architectural design, optimization objectives, normalization strategies, and sampling procedures, these artifacts tend to accumulate in the reconstruction error. Therefore, the residual provides a compact representation of generator-specific traces while suppressing semantic image content.

The motivation for employing a denoising autoencoder is rooted in forensic residual analysis. A denoising autoencoder trained on real images learns to reconstruct the dominant semantic content and natural image statistics while suppressing stochastic perturbations and reconstruction-inconsistent patterns. When a generated image is passed through the denoising autoencoder, the reconstructed output primarily preserves image content, whereas the residual image captures information that cannot be fully explained by the learned manifold of natural images. Prior forensic studies have shown that residual representations often reveal generator-specific artifacts and statistical inconsistencies that are less dependent on image semantics and more closely related to the image formation process. Consequently, the residual image acts as a content-suppressed representation that highlights subtle traces introduced during the image generation process, providing cues that are useful for attribution.

### 3.4.2 Frequency Aware Relation Module

The objective of this model is to explicitly incorporate frequency-domain features to enhance attribution accuracy by reducing the dependency on the content of the image. To achieve this, we leverage wavelet transform Mallat (1989), a powerful tool for frequency analysis, specifically employing 2D Discrete Wavelet Transform ($2D\ DWT$) to decompose input images into multiple frequency components. The conventional $2D\ DWT$ employs two filters: $L$ (low-pass filter), which retains low-frequency components, and $H$ (high-pass filter), which captures high-frequency details. Recent findings Miao et al. (2023) suggest that high-frequency features play a more critical role in face forgery detection than low-frequency ones. Therefore, we selectively retain only the $LH$ (low-high), $HL$ (high-low), and $HH$ (high-high) sub-bands for further processing, ensuring the extraction of discriminative frequency features. The feature extracted by $2D\ DWT$ can be formulated as:

$$W = DWT(X; LH, HL, HH) \tag{2}$$

where $X$ is the input image, $W$ is the extracted $2D\ DWT$ features which contain $LH$, $HL$ and $HH$ sub bands. If the size of the input image is $M \times N$, the size of each subband is $\frac{M}{2} \times \frac{N}{2}$ and each subband has the same number of channels as the input image (for $RGB$ image it is 3).

Multi-Head Self-Attention Mechanism: Once we have the $R$ (from denoising autoencoder) and $W$ (from frequency aware relation module), we upsample $LH$, $HL$ and $HH$ sub bands of $W$ using bicubic interpolation to match the dimension of $R$. Subsequently, $R$ is concatenated with upsampled $LH$, $HL$ and $HH$ sub bands of $W$ to form a representation given by:

$$J = \text{Concat}(R, LH, HL, HH) \in \mathbb{R}^{M \times N \times 4C} \tag{3}$$

where $M \times N$ is the spatial resolution and $C$ represents the channel dimension (here $C = 3$). We apply multi-head self-attention with four heads on the input feature map $J$ to refine the extracted features. The multi-head attention mechanism gives a refined feature map $Z$ that emphasizes attribution-relevant patterns and enhances the discriminative representation of the input features. The output $Z$ maintains the same spatial dimensions as $J$ but with enhanced global contextual information.

### 3.5 Feature Extraction Network

For attribution, a standard deep learning-based classifier can be trained on images generated from different generative models Yu et al. (2020). The main drawback of using a simple classifier is the drop in accuracy due to the presence of overlapping images as shown in Fig. 3 (a). While real images also exhibit overlapping features, classifiers can still perform well because they learn discriminative features from diverse datasets. However, in the case of generative model attribution, the feature overlap from structural and statistical similarities introduced by the generation process, leading to less distinctive patterns and making it harder for the classifier to learn decision boundaries.

To overcome this limitation, we incorporate a feature extraction network, which leverages supervised contrastive learning to enhance generalization and improve attribution performance. The feature extraction network is designed to generate discriminative embeddings that effectively capture the subtle traces left by different generative models. It is built upon a convolutional neural network ($CNN$) backbone, followed by a two-branch projection: a 128-dimensional projection head for contrastive loss computation and a 2048-dimensional classification head for final prediction. To train this network, we employ supervised contrastive learning Khosla et al. (2020), which improves feature robustness by explicitly modelling inter-class and intra-class relationships. In contrast to traditional cross-entropy loss, which only considers class labels for final classification, supervised contrastive learning forms contrastive pairs using label supervision during training. Given a batch of samples, the model pulls together representations of samples with the same class label (positive pairs) and pushes apart those from different classes (negative pairs). This is achieved via a contrastive loss function that operates on the cosine similarity between normalized feature embeddings.

We train feature extraction network with the feature map $Z$, extracted through the multi-head self-attention mechanism. Feature extraction network produces a 2048-dimensional feature embedding, which is subsequently reduced to a 128-dimensional feature embedding using a deep neural network. The 2048-dimensional feature embedding is called the classification head, which is used to train the multi-class classification network for attribution. The 128-dimensional feature embedding is called the projection head, which is used to calculate the supervised contrastive loss for training the feature extraction network. Using supervised contrastive loss results in a well-structured feature space where the same models are clustered together and dissimilar ones are distinctly separated, as shown in Fig. 3(b)

### 3.6 Multi-class Classification Network:

Our multi-class classification network is a deep neural network with fully connected layers. We trained this network using the 2048-dimensional feature embeddings generated from the classification head of feature extraction network. This multi-class classification network makes the final attribution of the generative models.

### 3.7 Loss Functions:

To optimize the entire pipeline, we integrate three loss functions:

- Multi-Head Attention Loss ($L_{MHA}$) promotes effective contextual feature extraction by aligning the input ($J_i$) and refined ($Z_i$) feature maps for batch of $N$ samples:

$$L_{MHA} = \frac{1}{N} \sum_{i=1}^{N} ||J_i - Z_i||^2 \tag{4}$$

The purpose of ($L_{MHA}$) is not to force an identity mapping, but to regularize the attention module so that it preserves attribution-relevant information while being jointly optimized with the supervised contrastive and classification losses. An identity mapping would not minimize the overall objective unless it also achieved optimal feature separability and attribution performance.

- Supervised Contrastive Loss ($L_{SCL}$) improves feature separability by bringing embeddings from the same generative model closer and pushing apart those from different ones:

$$L_{SCL} = \sum_{i \in I} \left( -\frac{1}{|P(i)|} \sum_{p \in P(i)} \log \left( \frac{\exp(f_i \cdot f_p/\tau)}{\sum_{a \in A(i)} \exp(f_i \cdot f_a/\tau)} \right) \right) \tag{5}$$

Here, $I$ is the set of all images in a mini-batch. $i \in I$ represents the index of an image in the mini-batch. $P(i) \equiv p \in A(i) : \tilde{y}_p = \tilde{y}_i$ denotes the set of positive samples ( sample generated by the same generative model) excluding $i$ itself. $|P(i)|$ is the number of such positive samples. $A(i) \equiv I \setminus \{i\}$ is the set of all other images in the batch. $f_i$ is the feature embedding and $\tau$ is the temperature scaling factor. The 128-dimensional projection head is utilized for contrastive loss computation during supervised training. Meanwhile, the 2048-dimensional feature embedding, referred to as the classification head, is used for multi-class classification.

- Cross-Entropy Loss ($L_{CE}$) guides the classification of the 2048-dimensional embedding using cross-entropy loss with *Adam* optimizer and a learning rate of 0.00003:

$$L_{CE} = -\sum_{i=1}^{N} y_i \log(\hat{y}_i) \tag{6}$$

where $y_i$ and $\hat{y}_i$ are the ground truth and predicted labels, respectively.

The overall loss is a weighted sum:

$$L_{Total} = \lambda_1 L_{MHA} + \lambda_2 L_{SCL} + \lambda_3 L_{CE} \tag{7}$$

This combination ensures that *GTA-Net* learns both discriminative and context-aware representations for reliable attribution of generated images.

## 4 Experimental Setup

In this section, we demonstrate the effectiveness of *GTA-Net* through comprehensive experiments. We performed all our experiments using the machine with *Intel Xeon CPU*, 32 *GB RAM*, and *Nvidia A*-6000 *GPU* with 48 *GB VRAM*. Training *GTA-Net* requires a diverse set of pre-trained generative models that can generate high-quality fake images. These models help us build a rich and diverse dataset to train and evaluate our network. We use 14 generative models in total: 8 *GAN*s, 2 diffusion models, 1 multimodal model, 1 autoregressive model, 1 flow-matching model, and 1 transformer-based model, which generate high-fidelity images for attribution tasks (detailed in Section 4.2). We selected these models because they collectively span the major families of modern generative modelling. These models differ in objective functions, sampling mechanisms and representation spaces, ensuring exposure to a wide spectrum of generative artifacts and improving the robustness and generalization of the classifier.

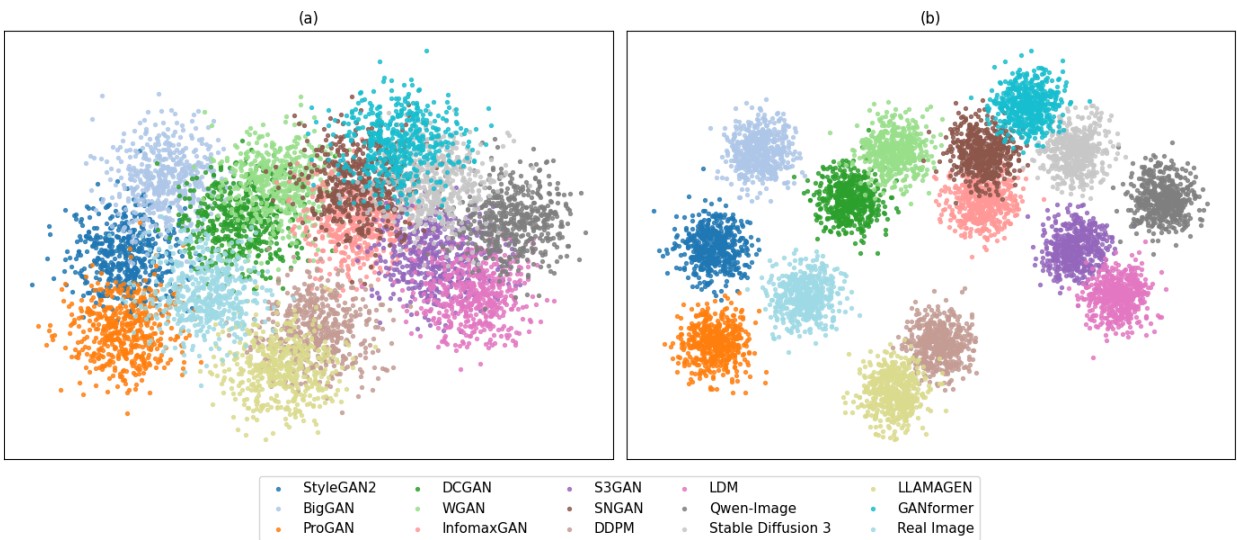

Figure 3: $t-SNE$ plot of feature embeddings of different generative models. In Fig. (a) the t-SNE plot is generated by passing images directly through the feature extraction network and the extracted features are visualized in the embedding space. In Fig. (b), the images are first processed by an abstractor network, which transforms them before being passed through the feature extraction network. The extracted features are then used for visualization.

## 4.1 Dataset and Model Architecture

We used $CelebA$ Liu et al. (2015), $LSUN\text{-}Cat$ Yu et al. (2015), and $MS\text{-}COCO$ Lin et al. (2014) datasets to train all the fourteen generative models, which are to be attributed based on their generated data. The $GAN$ models used for attribution are $StyleGAN2$ Karras et al. (2020), $BigGAN$ Brock et al. (2019), $ProGAN$ Karras et al. (2017), $DCGAN$ Radford et al. (2015), $WGAN$ Arjovsky et al. (2017), $InfomaxGAN$ Lee et al. (2021b), $S3GAN$ Lučić et al. (2019), and $SNGAN$ Miyato et al. (2018). For diffusion model, we use $DDPM$ Ho et al. (2020b), and $LDM$ Rombach et al. (2022). For transformer based we have used $GANformer$ Hudson & Zitnick (2021). At last, for multimodal diffusion we used $Qwen\ Image$ Wu et al. (2025), for the flow matching model we have used $Stable\ Diffusion$ 3 Esser et al. (2024) , and for autoregressive model we have used $LlamaGen$ Sun et al. (2024). We selected these generative models spanning the dominant paradigms of modern image synthesis to ensure architectural, objective-level, and representational diversity within the training space of $GTA\text{-}Net$. These models collectively represent distinct training objectives (min–max adversarial optimization vs. iterative denoising likelihood maximization), sampling strategies (single-pass generation vs. multi-step refinement vs flow matching), and feature inductive biases (convolutional locality, spectral normalization, style modulation, token generation, multimodality, and attention-based global modeling). By including representative architectures from each major generative family rather than multiple minor variants within a single family, we ensure that $GTA\text{-}Net$ learning discriminative representations rather than overfitting to superficial implementation details. This design choice provides maximal coverage of the generative modeling landscape while maintaining computational feasibility, and establishes a structured embedding space that supports robust nearest-class attribution for unseen models belonging to related generative families.

The apparent numerical imbalance between GAN-based and diffusion-based models in the training set is intentional rather than incidental. GANs represent a broader design space with significant architectural and objective-level variability (e.g., foundational CNN-GANs, spectral-normalized variants, large-scale class-conditional models, and style-modulated architectures), each introducing distinct inductive biases in feature synthesis and discriminator feedback dynamics. In contrast, diffusion-based models share a common proba-bilistic denoising formulation, where architectural variations primarily affect backbone implementation (e.g., U-Net vs. latent U-Net) rather than the underlying generative objective. Consequently, fewer diffusion

representatives are sufficient to capture the diffusion-family fingerprint, while multiple GAN variants are required to span the adversarial design space. Similarly, GANformer is included as a representative of transformer-augmented generative modeling, where attention replaces convolutional locality while preserving adversarial optimization. Thus, the training set is balanced at the level of generative paradigms rather than raw model counts, ensuring coverage of adversarial, likelihood-based, and attention-driven synthesis mechanisms without over-representing redundant objective formulations.

$GTA\text{-}Net$ is trained on the data generated by these generative models. The denoising autoencoder present in the abstractor network is a pre-trained model that is trained on the combination of $CelebA$ and $MS\text{-}COCO$ datasets. Its architecture is based on $CNN$ and consists of both an encoder and a decoder, each comprising 5 convolutional layers, excluding pooling and normalization layers. For feature extraction network, we adopt the encoder network and projection network used in Khosla et al. (2020). The encoder network, which utilizes $ResNet$-101 He et al. (2015) as its backbone, maps all the input samples into 2048-dimensional feature embeddings, ensuring robust representation learning. The projection network transforms feature embedding into a size of 128-dimension feature embedding by using a multi-layer perceptron with a single hidden layer of size 2048. The multi-class classification network consists of four fully connected layers with 1024, 512, 256, and 12 neurons, respectively. $ReLu$ activation is used in the intermediate layers, and $Softmax$ is in the final layer.

## 4.2 Training Details

To improve generalization, we train three different instances of each generative model on $CelebA$ by varying seeds and hyperparameters like learning rate, batch size, training steps, and initial model parameters. This introduce diversity while preserving the fundamental structure of each generative model. We choose three instances to strike a balance between diversity and computational efficiency, ensuring sufficient variability in training while keeping the dataset size manageable for effective learning. Once trained, each instance generates $5,000$ images, leading to $210,000$ fake images. We also include $40,000$ real $CelebA$ images, forming a dataset of $250,000$ samples. Training and validation sets were constructed at the image level while preserving class balance across all generative models. For each of the three training instances of every generative model, generated images were randomly partitioned using an 80:20 split, yielding approximately 200,000 training samples and 50,000 validation samples. Images generated by the fourth and fifth model instances used for cross-seed and cross-dataset evaluations were excluded entirely from training and validation and were reserved exclusively for testing. This protocol ensures that validation measures generalization on unseen images while cross-seed and cross-dataset experiments evaluate generalization to previously unseen model instances.

In the abstractor network, we use a pre-trained denoising autoencoder trained on real images from the combination of the $CelebA$ and $MS\text{-}COCO$ datasets. The autoencoder learns a generic natural-image manifold, allowing the residual image to emphasise deviations introduced by the generation process. We use mean absolute error as loss metric, $Adam$ optimizer, and set the learning rate at 0.0001. We employ mean absolute error rather than mean squared error to train the denoising autoencoder as mean squared error disproportionately penalizes large reconstruction errors and tends to produce overly smooth reconstructions that suppress subtle high-frequency artifacts. In contrast, mean absolute error provides a more robust optimization objective that preserves fine-grained residual structures and is less sensitive to outliers. Since GTA-Net relies on the residual image for attribution, preserving these weak reconstruction discrepancies is more important than achieving pixel-perfect reconstruction.

We resize the images to $128 \times 128$ for training and apply augmentation to the images (rotation, blurring, flipping, centre crop etc.). Training $GTA\text{-}Net$ involves training of abstractor network, feature extraction network and multi-class classification network in the pipeline. The training begins by passing the image through the denoising autoencoder and frequency-aware relation module to obtain the residual image $R$ and 2D $DWT$ feature $W$. Both $R$ and $W$ are concatenated and passed through a multi-head self-attention mechanism to obtain the refined feature map $Z$, as mentioned in the previous section. We compute multiple attention "heads" in parallel, each learning to focus on different types of patterns. The outputs of these attention heads are then merged into a single refined feature representation $Z$, which is used to train feature extraction network. We train the feature extraction network using a supervised contrastive loss, and pass

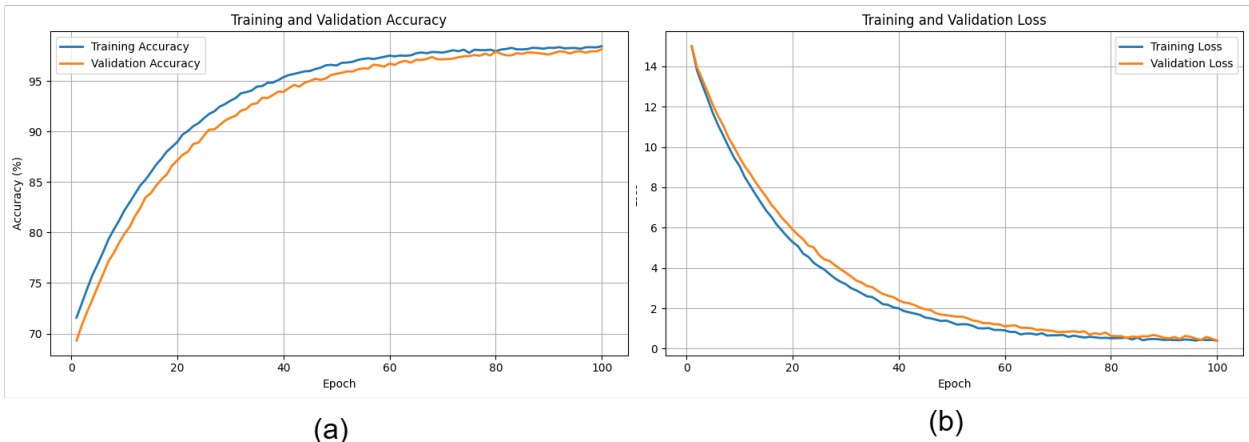

Figure 4: Training and validation accuracy of *GTA-Net* is shown in (a) and training and validation loss of *GTA-Net* is shown in (b)

the 2048-dimensional embedding into a multi-class classification network, enabling end-to-end training of *GTA-Net*.

## 5 Results

To assess the effectiveness of the proposed *GTA-Net* technique, we evaluate its performance in attribution.

### 5.1 Training and Validation Accuracy

To evaluate the effectiveness of *GTA-Net*, we use classification accuracy as the primary performance metric. The model demonstrates stable training, with loss convergence observed within 100 epochs, ensuring reliable optimization. On the validation set, *GTA-Net* achieves an impressive 98.16% average classification accuracy, highlighting its ability to accurately attribute generated images to their respective generative models. A comparative analysis of training and validation accuracy is illustrated in Fig. 4, showcasing the model's consistent performance across different stages of training. Additionally, we analyze the classification accuracy for individual generative models, providing a detailed breakdown in Table 1, further validating the robustness of our approach.

### 5.2 Feature Consistency and Separability

The abstractor network and supervised contrastive loss ($L_{SCL}$) in the feature extraction network enhance feature separability by ensuring that embeddings from the same generative model remain closely clustered while maintaining clear distinctions between different models. This facilitates more precise attribution. The *t-SNE* visualization in Fig. 3 demonstrates this effect. In Fig. 3 (a), overlapping clusters indicate weaker separation, whereas in Fig. 3 (b), the clusters are more distinct, reflecting improved feature representation. These results highlight the ability of *GTA-Net* to learn discriminative representations that improve separability between generative models."

### 5.3 Effect of Cross-seed and Cross-dataset:

So far, we have evaluated the performance of *GTA-Net* on images generated from the three instances of the generative models used for training. To assess the generalizability of *GTA-Net*, we trained the fourth instance for each generative model on the *CelebA* dataset using different hyperparameters and used it for cross-seed testing. We generated 5,000 images from the fourth instance of each generative model and tested *GTA-Net* for attribution. Previous work attempted a similar test but was only successful in attributing cross-seed

Table 1: Classification accuracy of GTA-Net for different generative models.

| Generative Model | Train Accuracy (%) | Validation Accuracy (%) |
|---|---|---|
| StyleGAN2 | 99.2 | 98.9 |
| BigGAN | 99.1 | 98.8 |
| ProGAN | 99.0 | 98.7 |
| DCGAN | 98.4 | 98.1 |
| WGAN | 98.0 | 97.5 |
| InfomaxGAN | 98.5 | 98.2 |
| S3GAN | 98.4 | 98.1 |
| SNGAN | 98.7 | 98.4 |
| DDPM | 98.0 | 97.0 |
| LDM | 98.3 | 98.0 |
| Qwen-Image | 98.5 | 98.2 |
| Stable Diffusion 3 | 97.8 | 97.5 |
| LLAMAGEN | 98.4 | 98.1 |
| GANformer | 98.7 | 98.4 |
| Real Data | 98.6 | 98.3 |
| **Average Accuracy** | **98.46** | **98.16** |

data generated from ProGAN Khosla et al. (2020). In contrast, our technique successfully attributed images from all generative models, as shown in the third column of Table 2.

Similarly, for cross-dataset testing, we trained a fifth instance for each generative model using the *LSUN-Cat* dataset instead of *CelebA*. Since the learned representations exhibit robustness to both model retraining and dataset shifts, *GTA-Net* maintains strong attribution performance when evaluated on images generated from models trained on different datasets. We generated $5,000$ images from the fifth instance of each generative model to perform attribution. The results, presented in the fourth column of Table 2, further validate the robustness of our technique.

Table 2: Attribution performance of GTA-Net under different settings: cross-seed, cross-dataset, and fine-tuning compared to the baseline model.

| Generative Model | Baseline | Cross-Seed | Cross-Dataset | Normal Fine-Tune | Cross Fine-Tune |
|---|---|---|---|---|---|
| StyleGAN2 | 98.9 | 98.4 | 98.1 | 98.3 | 97.9 |
| BigGAN | 98.8 | 98.3 | 98.0 | 98.2 | 97.8 |
| ProGAN | 98.7 | 98.2 | 97.9 | 98.1 | 97.7 |
| DCGAN | 98.1 | 97.8 | 97.5 | 97.7 | 97.3 |
| WGAN | 97.5 | 97.3 | 97.1 | 97.2 | 97.0 |
| InfomaxGAN | 98.2 | 97.7 | 97.4 | 97.6 | 97.2 |
| S3GAN | 98.1 | 97.6 | 97.3 | 97.5 | 97.1 |
| SNGAN | 98.4 | 97.9 | 97.7 | 97.8 | 97.4 |
| DDPM | 97.0 | 97.1 | 97.0 | 97.0 | 97.0 |
| LDM | 98.0 | 97.5 | 97.2 | 97.4 | 97.0 |
| Qwen-Image | 98.2 | 97.8 | 97.5 | 97.7 | 97.3 |
| Stable Diffusion 3 | 97.5 | 97.3 | 97.1 | 97.2 | 97.0 |
| LLAMAGEN | 98.1 | 97.7 | 97.4 | 97.6 | 97.2 |
| GANformer | 98.4 | 97.9 | 97.6 | 97.8 | 97.4 |
| Real Data | 98.3 | 98.5 | 98.5 | 98.5 | 98.5 |
| **Average Accuracy** | 98.16 | 97.80 | 97.56 | 97.71 | 97.32 |

### 5.4 Effect of Fine-tuning

Fine-tuning a generative model for additional epochs or steps alters its generation capability, leading to variations in the images produced for the same noise input. A robust attribution network should remain unaffected by fine-tuning as long as the underlying architecture and training procedure remains unchanged. To evaluate whether fine-tuning impacts *GTA-Net*'s attribution capability, we conducted the following experiment:

- We fine-tuned the fourth and fifth instances of each generative model on the *CelebA* dataset for 20 additional epochs.

- The fine-tuned fourth instance was labelled normal-fine-tune, while the fine-tuned fifth instance was labelled cross-fine-tune.

- We then generated $5,000$ images from each fine-tuned model and tested *GTA-Net* for attribution.

As shown in the last two columns of Table 2, *GTA-Net* achieved satisfactory results, demonstrating its robustness against fine-tuning.

### 5.5 Comparison

We compared our *GTA-Net* with existing deepfake attribution techniques of *LeveFreq* Frank et al. (2020), *AttNet* Yu et al. (2019), *DNA-Det* Yang et al. (2022), *Openworld-GAN* Girish et al. (2021), *POSE* Yang et al. (2023), *CPL* Sun et al. (2023), Handcrafted Li et al. (2024), ExDa Mu et al. (2025), Decoupled attention Zheng et al. (2025), Frequency Zhang et al. (2025), and MAID Zhu et al. (2025). Among these, *DNA-Det* and *POSE* are designed for cross-seed attribution, but it has been tested exclusively on *ProGAN*. The techniques, *LeveFreq*, *AttNet*, and Frequency focus on *GAN* attribution for models seen during training, meaning both training and testing data originate from the same generative model. *Openworld-GAN*, *POSE*, *CPL*, and Handcrafted Li et al. (2024) are used for open-set model attribution but achieves accuracy of around 70%. To ensure a fair comparison, we trained all the models using the same dataset used for training *GTA-Net*, following their respective experimental setups. The test results are shown in Table 3. From these results, it is clear that the existing attribution models except *LeveFreq* performed well on closed-set data. The cross-seed attribution results were not entirely satisfactory, as the attribution accuracy for *ProGAN* and *WGAN* was only slightly above 85%. Furthermore, a significant drop in accuracy was observed for *AttNet*, *LeveFreq*, *DNA-Det*, *Openworld-GAN*, *POSE*, Handcrafted, and Frequency when tested on the cross-seed data of other generative models. We also tested the existing techniques with the cross-dataset and fine-tuned data. The accuracy of the existing techniques dropped significantly, as shown in the last three columns of Table 3. This implies that *GTA-Net* outperforms the existing techniques in attribution under cross-seed, cross-dataset and fine-tuning scenarios.

### 5.6 Effect of Conditions on Generative Models

To further evaluate the attribution performance of *GTA-Net*, we trained conditional versions of *BigGAN*, *ProGAN*, *S3GAN*, *StyleGAN*2, *GANformer*, and *LDM* on the *MS-COCO* dataset. During the training of *GTA-Net*, we used images generated from their unconditional counterparts. Since the underlying architectures remain unchanged, with only minor modifications in the training procedure to accommodate conditioning, this experiment allows us to assess whether *GTA-Net* is robust to such variations. To conduct this evaluation, we generated $5,000$ images from each conditional generative model and tested their attribution using *GTA-Net*. If *GTA-Net* successfully attributes these images, it suggests that the learned representations remain informative despite moderate changes in training procedures rather than being overly sensitive to slight changes in training methodology. The results of this evaluation are presented in Table 4.

### 5.7 Evaluation on Generative Model in the Wild

To assess the generalization capability of *GTA-Net* in the wild, we perform attribution on generative models that were not included in the training phase. Rather than assigning images from unknown generative models

Table 3: Comparison of *GTA-Net* with existing deepfake attribution techniques in different attribution scenarios. The table shows accuracies obtained under closed-set, cross-seed, cross-dataset, and fine-tuned scenarios.

| Technique | Closed-Set | Cross-Seed | Cross-Dataset | Normal Fine-Tune | Cross Fine-Tune |
|---|---|---|---|---|---|
| LeveFreq | 72.1 | 52.4 | 48.6 | 61.3 | 42.7 |
| AttNet | 84.8 | 61.9 | 37.2 | 64.1 | 35.4 |
| DNA-Det | 89.1 | 72.5 | 67.1 | 79.5 | 70.2 |
| Openworld-GAN | 81.8 | 68.7 | 59.2 | 70.3 | 58.1 |
| POSE | 90.3 | 68.7 | 84.2 | 86.7 | 82.4 |
| CPL | 94.1 | 84.5 | 80.1 | 89.4 | 79.2 |
| Handcrafted | 97.3 | 86.2 | 78.2 | 81.3 | 78.1 |
| ExDa | 77.8 | 73.2 | 61.1 | 82.6 | 68.7 |
| Decoupled Attention | 95.25 | 69.7 | 88.2 | 81.3 | 78.1 |
| Frequency | 84.6 | 79.9 | 76.7 | 92.8 | 81.4 |
| MAID | 89.1 | 81.8 | 68.4 | 83.7 | 76.9 |
| **GTA-Net** | **98.16** | **97.80** | **97.56** | **97.71** | **97.32** |

Table 4: Performance of *GTA-Net* on conditional generative models. We compared our results to a baseline model trained on an unconditional generative model.

| Generative Model | Baseline (%) | Conditional Model (%) |
|---|---|---|
| BigGAN | 99.6 | 97.3 |
| ProGAN | 99.7 | 96.4 |
| S3GAN | 98.3 | 95.2 |
| StyleGAN2 | 99.6 | 96.9 |
| GANformer | 98.1 | 95.7 |
| LDM | 97.5 | 93.0 |
| **Average Accuracy** | 98.9 | 95.5 |

that are not included in *GTA-Net*'s training (also do not belong to real image class) to a separate class, we instead attribute them to the closest class in the *GTA-Net* adapting a nearest-class attribution strategy. This technique ensures that even unseen generative models are mapped to the most similar model within the existing options and the ambiguity around threshold sensitivity associated with open-set rejection can be avoided. We evaluate attribution performance on three different categories of generative models. The first category includes generative models that synthesize images directly from noise, optionally conditioned on text prompts, without requiring an input image. This category covers *BEGAN* Berthelot et al. (2017), *CramerGAN* Bellemare et al. (2017), *MMDGAN* Li et al. (2017), *ProjectedGAN* Sauer et al. (2021), *StyleGAN* Karras (2019), *StyleGAN3* Karras et al. (2021a), *TransGAN* Jiang et al. (2021), *ViTGAN* Lee et al. (2021a), *Stable Diffusion* Rombach et al. (2022), *GLIDE* Nichol et al. (2021), *Taming Transformer* Esser et al. (2021), *Self-Attention GAN* (*SAGAN*) Zhang et al. (2019), *Latent Consistency Model* (*LCM*) Bai et al. (2023), *CIPS* Anokhin et al. (2021), *DiffusionGAN* Wang et al. (2022), *Denoising Diffusion GAN* Xiao et al. (2021), *DiT* Peebles & Xie (2023), *U-ViT* Bao et al. (2023), *VQ-Diffusion* Gu et al. (2022), *DeepFloyd IF* at StabilityAI (2023), *Stable Diffusion XL* Podell et al. (2024), *PixArt-α* Chen et al. (2024), *Stable Diffusion* 3.5 Esser et al. (2024), *FLUX*.1 Labs (2024), *Lumina-T2X* Gao et al. (2024), *SANA* Xie et al. (2024), *Nano Banana* Team et al. (2023), *PixelSNAIL* Chen et al. (2018), and *Imagen* Saharia et al. (2022b). The second category consists of generative models that require an input image, mask, or semantic layout as conditioning information. These models perform image-to-image translation, attribute manipulation, semantic synthesis, or inpainting. This includes *Pix2Pix* Isola et al. (2017), *StarGAN* Choi et al. (2018), *Palette* Saharia et al. (2022a) , *CycleGAN* Zhu et al. (2017), *Pix2Pixhd* Wang et al. (2018), *LaMa* Suvorov et al. (2022), *MAT* Li et al. (2022), *GauGAN* Park et al. (2019), *AttGAN* He et al. (2019), *STGAN* Liu et al. (2019), *MaskGIT* Chang et al. (2022), *Nano Banana* Team et al. (2023)

and *RelGAN* Wu et al. (2019). The third category consists of modified variants of *ProGAN*, *StyleGAN*, *SNGAN*, *GANformer* and *DDPM*, ensuring that *GTA-Net* is tested on a wide range of architectures, as presented in Table 5.

Table 5: Architectural and Loss Function Modifications in Generative Models

| Model | Architectural Modifications | Loss Function Modifications |
|---|---|---|
| ProGAN | Progressive growing of layers; modified upsampling strategy | Hinge loss |
| StyleGAN | Mapping network; style modulation (AdaIN); noise injection | R1 regularization; perceptual loss |
| SNGAN | Spectral normalization applied to discriminator layers | Hinge or Wasserstein adversarial loss |
| GANformer | Transformer-based self-attention layers in generator and discriminator | Hinge loss with attention regularization |
| DDPM | U-Net backbone with residual blocks and time-step embeddings | Noise prediction loss (L2 objective) |

These models span both *GAN*-based, transformer-based, flow-based, autoregressive and diffusion-based architectures, ensuring a diverse evaluation set. We generate a total of 2000 images from each model to perform attribution analysis. *GTA-Net* is trained to attribute images to their source generative models by learning discriminative representations that remain informative across a range of model variations. It successfully attributes images from unseen generative models to the closest trained counterpart, as shown in Table 6, Table 7, and Table 8 . When performing attribution on known models, the confidence score is generally above 90%. However, for unseen generative models, we observe a lower average confidence score. Despite this, *GTA-Net* maintains high attribution accuracy, effectively mapping unseen models to their most similar trained counterparts.

From Table 6, the following observations can be made:

- **GAN-based models cluster according to adversarial training dynamics:** CNN-based GAN variants are consistently attributed to their closest adversarial counterparts with moderate-to-high confidence. This indicates that discriminator-driven optimization and convolutional inductive biases introduce stable fingerprints that persist across architectural variants.

- **Transformer-GANs preserve family-level characteristics despite architectural shifts:** *TransGAN* and *ViTGAN* are correctly mapped to *GANformer* with lower confidence than CNN-GANs. This suggests that attention-based generation modifies low-level synthesis artifacts while retaining the underlying adversarial optimization model.

- **Diffusion objectives dominate attribution behavior:** Pixel-space, latent-space, and transformer-based diffusion models are consistently mapped to diffusion-based classes with high confidence and accuracy. This indicates that the iterative denoising process introduces stronger and more persistent forensic traces than backbone-specific architectural choices.

- **Latent diffusion models exhibit the strongest attribution consistency:** *Stable Diffusion*, *Stable Diffusion XL*, and related latent diffusion models achieve among the highest confidence and attribution accuracy. The latent compression and reconstruction stages appear to reinforce diffusion-specific fingerprints, producing highly stable discriminative representations.

- **Flow-matching models form a highly coherent attribution cluster:** *FLUX*.1, *Lumina-T2X, SANA*, and *Stable Diffusion* 3.5 are consistently attributed to *Stable Diffusion* 3 with the highest overall confidence. This suggests that transport-based generation objectives induce highly consistent family-level fingerprints that generalize across different implementations.

- **Autoregressive image generators produce distinct generative traces:** *PixelSNAIL* is accurately attributed to *LLAMAGEN*, indicating that sequential token prediction introduces characteristic statistical dependencies that are clearly separable from both adversarial and diffusion-based generation processes.

Table 6: Nearest-Class Attribution on Unseen Noise-Based / Full-Synthesis Models

| Model Type | Model Name | Attributed Model | Confidence (%) | Accuracy (%) |
|---|---|---|---|---|
| CNN-GAN | BEGAN | DCGAN | 74.6 | 82.1 |
| CNN-GAN | CramerGAN | WGAN | 76.8 | 84.3 |
| CNN-GAN | MMDGAN | WGAN | 75.2 | 83.5 |
| Stabilized GAN | ProjectedGAN | SNGAN | 81.4 | 88.6 |
| Style-Based GAN | StyleGAN | StyleGAN2 | 87.2 | 93.4 |
| Style-Based GAN | StyleGAN3 | StyleGAN2 | 88.1 | 94.1 |
| Self-Attention GAN | SAGAN | BigGAN | 84.6 | 89.2 |
| Transformer-GAN | TransGAN | GANformer | 53.8 | 83.4 |
| Transformer-GAN | ViTGAN | GANformer | 51.9 | 82.7 |
| Implicit Generator | CIPS | StyleGAN2 | 41.3 | 71.5 |
| Pixel Diffusion | GLIDE | DDPM | 81.7 | 89.4 |
| Pixel Diffusion | DeepFloyd IF | DDPM | 84.2 | 91.3 |
| Latent Diffusion | Stable Diffusion | LDM | 89.6 | 95.2 |
| Latent Diffusion | Stable Diffusion XL | LDM | 91.4 | 96.1 |
| Latent Diffusion | LCM | LDM | 87.8 | 93.7 |
| Transformer Diffusion | DiT | Qwen-Image | 76.4 | 90.8 |
| Transformer Diffusion | U-ViT | Qwen-Image | 74.8 | 89.5 |
| Transformer Diffusion | PixArt-$\alpha$ | Qwen-Image | 83.5 | 92.6 |
| Transformer Diffusion | Nano Banana | Qwen-Image | 69.2 | 90.1 |
| Discrete Diffusion | VQ-Diffusion | Qwen-Image | 78.6 | 87.8 |
| Text Diffusion | Imagen | Qwen-Image | 86.9 | 93.4 |
| Flow Matching | FLUX.1 | Stable Diffusion 3 | 92.7 | 96.8 |
| Flow Matching | Lumina-T2X | Stable Diffusion 3 | 90.8 | 95.3 |
| Flow Matching | SANA | Stable Diffusion 3 | 89.4 | 94.6 |
| Flow Matching | Stable Diffusion 3.5 | Stable Diffusion 3 | 93.6 | 97.2 |
| Hybrid Diff-GAN | DiffusionGAN | DDPM | 79.3 | 87.6 |
| Hybrid Diff-GAN | Denoising Diffusion GAN | DDPM | 78.5 | 86.9 |
| Autoregressive | PixelSNAIL | LLAMAGEN | 88.3 | 94.4 |
| VQ + Transformer | Taming Transformer | LLAMAGEN | 82.4 | 89.8 |

- **Implicit generators exhibit the weakest attribution signals:** *CIPS* achieves the lowest confidence and attribution accuracy among all evaluated models. The absence of convolutional feature hierarchies and iterative denoising mechanisms reduces the similarity of its generated artifacts to the families represented during training.

- **Hybrid models are primarily grouped according to their dominant generative objective:** Models such as *DiffusionGAN*, *Denoising Diffusion GAN*, and *Taming Transformer* are attributed to the closest family corresponding to their primary generation mechanism, suggesting that optimization objectives contribute more strongly to attribution than secondary architectural components.

- **Confidence correlates with generative-family similarity:** Models belonging to well-represented families exhibit higher confidence scores, whereas architectures that deviate substan-

tially from the training distribution show reduced confidence. This indicates that confidence acts as a proxy for proximity within the learned generative-family embedding space.

Table 7: Nearest-Class Attribution on Unseen Conditional / Image-to-Image Models

| Tested Model | Attributed Model | Confidence (%) | Accuracy (%) |
|---|---|---|---|
| Pix2Pix | SNGAN | 57.5 | 64.9 |
| StarGAN | SNGAN | 58.1 | 65.7 |
| CycleGAN | DCGAN | 53.4 | 60.2 |
| Pix2PixHD | ProGAN | 60.3 | 67.4 |
| GauGAN | StyleGAN2 | 62.7 | 69.6 |
| AttGAN | SNGAN | 56.9 | 63.8 |
| STGAN | SNGAN | 55.3 | 62.4 |
| RelGAN | SNGAN | 57.6 | 64.5 |
| Palette | DDPM | 59.6 | 66.8 |
| LaMa | LDM | 81.2 | 88.1 |
| MAT | GANformer | 59.8 | 66.5 |
| Nano Banana | Qwen-Image | 54.6 | 81.3 |
| MaskGit | GANFormer | 83.2 | 89.8 |

Table 7 highlights that conditional and image-to-image generative models exhibit comparatively lower attribution confidence and accuracy than full-synthesis models, suggesting that conditioning signals partially obscure intrinsic generative artifacts. Most adversarial translation frameworks are consistently mapped to $SNGAN$ with moderate confidence and accuracy, indicating that despite task-specific conditioning mechanisms, their core adversarial training pipeline and convolutional generator–discriminator structure dominate the learned artifact space. $CycleGAN$ aligns with $DCGAN$, reflecting its foundational CNN-based adversarial formulation without advanced normalization or style modulation. $Pix2PixHD$ maps to $ProGAN$, likely due to its progressive high-resolution synthesis strategy, while $GauGAN$ aligns with $StyleGAN2$, suggesting that semantic layout–conditioned generation shares structural similarities with style-modulated feature injection. Diffusion-based conditional models demonstrate clearer separation: Palette is attributed to $DDPM$, indicating that pixel-space diffusion artifacts persist even under inpainting tasks, whereas $LaMa$ achieves substantially higher confidence and accuracy when mapped to $LDM$, implying that latent-space compression and denoising introduce strong, consistent discriminative representations that survive conditional masking. $MAT$, which integrates transformer-based components, is attributed to $GANformer$ with moderate performance, suggesting that attention mechanisms influence artifact similarity but remain secondary to the dominant training objective. Overall, these results indicate that in conditional generation settings, the conditioning pathway introduces variability that attenuates artifact strength, yet the underlying generative objective remains the primary factor governing nearest-class attribution behavior in $GTA\text{-}Net$.

Table 8: Nearest-Class Attribution on Architectural Variants (Category 3)

| Tested Model | Attributed Model | Confidence (%) | Accuracy (%) |
|---|---|---|---|
| ProGAN (modified) | ProGAN | 96.1 | 93.8 |
| StyleGAN (modified) | StyleGAN2 | 95.4 | 92.9 |
| SNGAN (modified) | SNGAN | 94.6 | 91.5 |
| GANformer (modified) | GANformer | 96.8 | 94.3 |
| DDPM (modified) | DDPM | 95.7 | 92.4 |

Table 8 demonstrates that $GTA\text{-}Net$ exhibits strong robustness to architectural and loss-function perturbations within the same generative family. All modified variants are attributed to their corresponding base architectures with very high confidence and accuracy, indicating that minor structural changes, normalization strategies, or objective refinements do not significantly alter the underlying generative fingerprint. These results indicate that GTA-Net remains robust under moderate architectural and objective-level modifications

within the same model family. The high attribution consistency suggests that the learned representations are not overly sensitive to implementation-level changes. In particular, the high attribution consistency for modified $ProGAN$ and $StyleGAN$ variants implies that progressive growing and style-modulated synthesis impose persistent structural biases in feature statistics. Similarly, the strong alignment of modified $SNGAN$ and $GANformer$ variants confirms that spectral normalization and transformer-based attention introduce family-specific artifact patterns that remain invariant under moderate architectural adjustments. The diffusion variant also preserves its discriminative attribution representations, indicating that iterative denoising dynamics generate stable, process-driven artifacts resistant to architectural tuning. Overall, these results validate that $GTA\text{-}Net$ learns generative-family representations that are invariant to intra-family modifications, reinforcing its suitability for robust real-world attribution where exact implementation details may vary.

These results collectively demonstrate that both training objectives and architectural design choices fundamentally shape the attribution behavior of $GTA\text{-}Net$. Across all categories, diffusion-based models exhibit the strongest and most consistent attribution confidence, indicating that iterative denoising and latent compression introduce stable, process-driven artifacts. In contrast, adversarial CNN-based GANs cluster according to shared inductive biases and optimization objectives, while transformer-based and implicit generators show comparatively lower confidence due to architectural divergence from convolutional priors. Conditional image-to-image models further reveal that conditioning pathways attenuate artifact strength, yet the underlying generative objective remains the dominant factor governing attribution similarity. The confidence score therefore serves not only as a measure of classification certainty but also as an indicator of generative-family alignment in the learned embedding space. High confidence reflects strong discriminative representations that remain informative across a range of model variations, whereas lower confidence may signal architectural novelty, hybrid training paradigms, or weakened artifact persistence. Importantly, the high intra-family attribution accuracy observed for architectural variants confirms that $GTA\text{-}Net$ captures stable generative priors rather than overfitting to specific implementations. Overall, these findings suggest that $GTA\text{-}Net$ generalizes beyond closed-set recognition and instead learns structured, objective-level generative representations, enabling robust nearest-class attribution for unseen models in realistic forensic scenarios.

## 5.8 Ablation Study

To quantify the importance of each module within $GTA\text{-}Net$, we conduct an ablation study by removing different components and training the model accordingly. The results, summarized in Table 9, reveal that:

- Removing the Multi-Head Self-Attention module decreases accuracy by 19%, confirming its role in effective feature refinement. Without this module, the model struggles to refine distinguishing features, leading to increased misclassifications, especially in cross-dataset and fine-tuned scenarios.

- Excluding the Supervised Contrastive Learning module reduces feature separability, as evidenced by overlapping clusters in the $t\text{-}SNE$ visualization. This suggests that contrastive learning plays a key role in enhancing representation learning, ensuring robustness even when models undergo retraining with different seeds or datasets.

- Without the Denoising Autoencoder, model performance degrades by 18%, highlighting its contribution to feature extraction. The autoencoder helps in mitigating variations introduced by different training conditions, ensuring that $GTA\text{-}Net$ remains effective in identifying the underlying discriminative representations despite changes in model initialization, loss functions, or dataset variations.

To further evaluate the sensitivity of $GTA\text{-}Net$ to input resolution, we trained the attribution pipeline at multiple image scales ($128 \times 128$, $256 \times 256$, $512 \times 512$, and $1024 \times 1024$) as shown in Table 10. While higher resolutions provide increased spatial detail, the attribution performance exhibits only marginal improvement beyond $128 \times 128$. This suggests that generative fingerprints exploited by $GTA\text{-}Net$ are primarily encoded in mid- to low-frequency statistical patterns rather than high-frequency pixel-level detail. Diffusion artifacts, convolutional inductive biases, and adversarial texture inconsistencies appear to be resolution-invariant once sufficient spatial structure is preserved. Consequently, scaling the input resolution increases computational

Table 9: Ablation study results for GTA-Net.

| Model Configuration | Accuracy (%) |
|---|---|
| Full GTA-Net | 98.8 |
| Without Multi-Head Attention | 81.2 |
| Without Supervised Contrastive Loss | 67.4 |
| Without Denoising Autoencoder | 82.3 |

cost (memory footprint and training time grow quadratically with spatial dimension) without yielding statistically significant gains in attribution accuracy. These findings indicate that $128 \times 128$ provides an optimal trade-off between computational efficiency and forensic discriminability, confirming that *GTA-Net* captures discriminative representations rather than resolution-dependent visual cues.

Table 10: Effect of Input Resolution on GTA-Net Attribution Performance

| Resolution | Accuracy (%) | Confidence (%) | Cost |
|---|---|---|---|
| $128 \times 128$ | 98.8 | 88.4 | $1\times$ |
| $256 \times 256$ | 98.9 | 89.1 | $4\times$ |
| $512 \times 512$ | 97.3 | 89.5 | $16\times$ |
| $1024 \times 1024$ | 98.9 | 89.7 | $64\times$ |

## 6 Discussion

Unlike synthetic images generated directly from a trained model, real-world manipulations sometimes involve edits such as face swaps, deepfake augmentations, and attribute manipulation techniques that do not follow a standard generative process. To validate the generalizability of our method in real-world scenarios, we evaluated *GTA-Net* on the *ForgeryNet* dataset He et al. (2021), which comprises 15 distinct manipulation techniques based on generative models. We randomly sampled $2,000$ manipulated images from each technique and performed attribution. Unlike traditional generative models that synthesize images from scratch, *ForgeryNet* primarily involves manipulation of existing images, such as identity swapping, expression transfer, and facial reenactment. As a result, the confidence scores for attribution were slightly lower due to the subtler generative traces. Nonetheless, the attribution accuracy remained strong, despite *GTA-Net* being trained exclusively on images fully generated by *GAN*s or diffusion models, without exposure to manipulated images during training which can be seen in Table 11. Many face manipulation pipelines involve multiple modules such as encoders, facial landmark extractors, motion transfer networks, and domain adaptation layers. However, our attribution focuses on the generative component, such as the backbone of the technique, the impact of the loss function, preprocessing/postprocessing modules, and training strategies, which jointly influence the final output. These factors may obscure the dominant generative representations, making precise attribution more challenging despite the model's overall strong performance. Still, this demonstrates the robustness of *GTA-Net* in detecting generative fingerprints even in complex, post-processed manipulations.

### 6.1 Analysis of Potential Confounders and Attribution Cues

A potential concern is that the high attribution accuracy achieved by GTA-Net may arise from semantic image content or dataset-specific artefacts rather than the discriminative representations that remain informative across a range of model variations. We therefore evaluate whether the learned representations are primarily organized according to generator identity or image content. First, the embedding-space visualization shown in Fig. 3 provides qualitative evidence that GTA-Net clusters images according to their source generator. Despite substantial visual similarity between images produced by different models, the learned representations form compact intra-generator clusters and remain well separated across generator families. This behavior suggests that the feature extractor captures generator-dependent characteristics rather than semantic image structures. Second, the cross-dataset evaluation presented in Table 2 provides quantita-

Table 11: Attribution results of GTA-Net on ForgeryNet.

| Manipulation Technique | Attributed Model | Confidence | Accuracy |
|---|---|---|---|
| FaceShifter | StyleGAN2 | 71.2% | 86.5% |
| FS-GAN | ProGAN | 59.8% | 80.2% |
| DeepFakes | DCGAN | 54.1% | 77.0% |
| BlendFace | StyleGAN2 | 68.9% | 85.0% |
| MMReplacement | WGAN | 51.6% | 75.3% |
| DeepFakes-StarGAN | SNGAN | 48.3% | 78.9% |
| Talking Head Video | DDPM | 60.4% | 80.5% |
| ATVG-Net | DDPM | 62.7% | 82.0% |
| StarGAN-BlendFace | SNGAN | 46.2% | 84.1% |
| First Order Motion | DDPM | 55.9% | 80.6% |
| StyleGAN2 | StyleGAN2 | 91.7% | 90.2% |
| MaskGAN | DCGAN | 56.0% | 76.5% |
| StarGAN2 | SNGAN | 67.4% | 85.1% |
| SC-FEGAN | StyleGAN2 | 70.6% | 86.9% |
| DiscoFaceGAN | ProGAN | 52.7% | 78.0% |

Table 12: Prompt-controlled attribution experiment. Images were generated from identical prompts using different prompt-conditioned generative models.

| Generative Model | Attribution Accuracy (%) |
|---|---|
| Qwen-Image | 98.4 |
| Stable Diffusion 3 | 97.9 |
| LDM | 97.6 |
| LlamaGen | 98.1 |
| Average | 98.0 |

tive evidence against dataset-specific memorization. GTA-Net is trained on images generated from models trained on CelebA and evaluated on images produced by the same architectures trained on LSUN-Cat. Since the semantic distribution of LSUN-Cat differs substantially from CelebA, successful attribution requires representations that generalize beyond the content distribution observed during training. GTA-Net maintains an average attribution accuracy of 97.71% under this setting, indicating that the learned features remain stable across significant changes in image content and training data distributions.

To further isolate the influence of semantic content, we conducted a prompt-controlled attribution experiment using prompt-conditioned generative models. Specifically, we constructed 50 highly constrained facial prompts describing frontal passport-style portrait images with fixed pose, viewpoint, background, and lighting conditions. Example prompts include: 'A frontal passport-style photograph of a young woman with black hair, neutral facial expression, white background, studio lighting' and 'A frontal passport-style photograph of a middle-aged man wearing rectangular eyeglasses, neutral facial expression, white background, studio lighting.' Each prompt was used to generate one image from Qwen-Image, Stable Diffusion 3, LDM, and LlamaGen, resulting in a total of 200 generated images. Because all generators received identical prompts and were constrained to produce highly similar portrait images, semantic variability between generators was substantially reduced. The attribution results are reported in Table 12. GTA-Net maintains an average attribution accuracy of 98.0% across all evaluated models despite the strong semantic similarity between generated images. If the attribution decision were primarily driven by image content, performance would be expected to degrade because images corresponding to the same prompt share nearly identical facial attributes, pose, illumination conditions, and scene composition. Instead, GTA-Net continues to attribute images correctly, indicating that the learned representations are not dominated by semantic content.

These behaviours can be explained by the design of GTA-Net. Prior to classification, the input image is transformed into a residual-frequency representation through the denoising autoencoder and wavelet decomposition modules. These representations suppress high-level semantic information while preserving subtle

artifacts introduced by the image synthesis process. The subsequent attention mechanism and supervised contrastive learning objective further encourage clustering according to generator-specific characteristics rather than image appearance. Consequently, GTA-Net learns stable discriminative representations associated with the underlying generation process, enabling robust attribution even when semantic content is explicitly controlled. Collectively, the t-SNE visualization, cross-dataset evaluation, and prompt-controlled attribution experiment provide strong evidence that GTA-Net primarily exploits generator-specific fingerprints rather than content-dependent cues or dataset-specific confounders.

# 7 Conclusion

In this paper, we proposed *GTA-Net* for attributing the generative model-based images to their original source generative model. To achieve this, we have developed a technique that can extract traces of discriminative representations from the generated images. *GTA-Net* filters out data-dependent features, allowing the feature extraction network to focus or concentrate on characteristics unique to the generative model such as its architecture and training process. To demonstrate the generalization capability of *GTA-Net*, we evaluated it across 15 different generative models. Our results show that *GTA-Net* successfully attributes generated images to their corresponding generative models, even when the images originate from models that were unseen during training. We also compared our approach with existing attribution methods to demonstrate the effectiveness of *GTA-Net*. Looking ahead, we plan to expand our technique to attribute images that are modified through real-world editing with higher accuracy. This will enable *GTA-Net* to attribute both synthesized and manipulated images to their respective sources.

# 8 Broader Impact Concerns

*GTA-Net* is intended to support forensic analysis, content provenance, copyright protection, and accountability in the use of generative models. By attributing generated images to their likely source models, such systems may assist investigators, platform operators, and researchers in understanding the origins of synthetic media and mitigating malicious use of generative AI.

However, attribution systems also present potential risks and limitations. First, attribution predictions are probabilistic rather than definitive proof of origin. Incorrect attribution may occur when a generated image originates from a previously unseen model, a heavily modified model, or a generation pipeline that differs substantially from those represented during training. Consequently, attribution results should be treated as supporting evidence rather than conclusive forensic evidence.

Second, in legal, regulatory, or forensic settings, overreliance on automated attribution systems could lead to incorrect conclusions regarding responsibility or ownership. GTA-Net is designed as a decision-support tool and should be used alongside additional technical, contextual, and investigative evidence rather than as the sole basis for attribution claims.

Third, malicious actors may attempt to evade attribution by modifying model architectures, training procedures, post-processing generated outputs, or intentionally designing attribution-resistant generation pipelines. Continued research is therefore necessary to evaluate the robustness of attribution systems under adversarial conditions.

Finally, while GTA-Net aims to improve transparency and accountability, it is important to recognize that generative AI technologies continue to evolve rapidly. Future generative paradigms may exhibit characteristics not represented in the current training distribution, potentially affecting attribution reliability. Users should therefore interpret attribution outputs with appropriate caution and awareness of these limitations.

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
