# OpenReview forum: "Generative Trace Attribution Network"
_TMLR — Rejected by TMLR_

### Review · Reviewer_WKZ3 · 2026-04-17

**Summary Of Contributions:**

The paper proposes a method for attributing deep fakes to a particular model or model class.
The main issue with prior methods is that they degrade in performance when small changes are made to a model such as different random seeds, different datasets, or finetuning.
The proposed approach uses a staged approach for extracting residuals and high-frequency details from images, imposing a contrastive learning objective on the representations and then clasifying based on the latent representations.

**Strengths**
- Architecture seems carefully designed to handle this specific classification task, in particular the preprocessing and contrastive objective seem like non-obvious components.
- The cross-seed, cross-dataset, finetune evaluations were interesting.
- Simple ablation study to check that all components are necessary.
- Strong empirical results compared to baselines.

**Weaknesses**
- Focuses on GAN-based models instead of more common diffusion-based models and also does not consider autoregressive models like VAR or similar. Justification for this distinction is made but unconvincing. This most appropriate models for deep fakes are diffusion models, flow-matching models and autoregressive generative models, which are the standard now.

- Parts of the methodology are not particularly well-justified beyond empirical support. In some cases, claims are made about what each part is doing before the empirical results are given. The claims should be after empirical results and posed as design decisions but overall the claims about "what" the part is doing should be more careful. It would be better if you started with the simple design and then showed through a series of small experiments why you needed to add the components.

- Organization is not the best. The comparison results should be first in the experiments and the main one to be highlighted. The section on just analyzing the current approach could be significantly shortened and simplified with details pushed to the appendix as without the context of other methods it's hard to understand or intepret.


**Questions**

- Why not use well-known and open-source pretrained models for training? Why train your own models?

- I don't understand the Multi-Head Attention Loss: Why wouldn't the identity function just work? Why do you need this loss?

**Audience:**

Yes

**Audience Explanation:**

Deep fake attribution is an interesting ML problem. I'm not 100% sure how often this will be important but I think it satisfies this requirement.

**Broader Impact Concerns:**

The authors should add a Broader Impact statement about how their system could be used or misused to attribute or wrongly attribute fakes to different sources. Since forensic evidence is used as an example, considering a forensic litigation case should be addressed. While I don't see any immediate major concerns, I think these and other potential impacts should be discussed.

**Claims And Evidence:**

No

**Claims Explanation:**

* The claim that the proposed method learns "architecture-level signatures" which is claimed throughout the paper, does not have ample evidence. Robustness is shown but the claimed reason for robustness "architecture-level signatures" is not supported. It could be phrased as a conjecture but would need explicit evidence for the reason. Several quotes below are related to this overclaiming for the "reason" for robustness.
    * "By learning discriminative, architecture-level signatures, GT A-N et achieves superior performance in both closed-world and open-world settings, offering reliable attribution even under black-box constraints and with photorealistic inputs."
    * "For previously unseen models that are not part of the training set, GT A-N et performs attribution based on architectural similarities, training pipeline, and loss functions."
    * "The multi-head attention mechanism gives a refined feature map Z, which is a representation that is specific to a given architecture and training pipeline".
    * "However, in the case of generative model attribution, the feature overlap from structural and statistical similarities introduced by the generation process, leading to less distinctive patterns and making it harder for the classifier to learn decision boundaries."

* "Training GT A-N et requires a diverse set of pre-trained generative models that can generate high-quality fake images. These models help us build a rich and diverse dataset to train and evaluate our network. We use eleven generative models in total, eight GAN s, 2 diffusion models, and one transformer based which generate high-fidelity images for attribution tasks (detailed in Section 4.2)." - The claim of rich and diverse is not validated since most generative models these days are diffusion models and most generative modles use transformer architectures.
    * "We selected these generative models spanning the dominant paradigms of modern image synthesis to ensure architectural, objective-level, and representational diversity within the training space of GT A-N et."
    * "By including representative architectures from each major generative family rather than multiple minor variants within a single family, we ensure that GT A-N et learns objective-level and architecture-level fingerprints rather than overfitting to superficial implementation details."
    * Justification for imbalance is poor. Almost all currently used models are diffusion or autoregressive (VAR) and almost all are transformer based.

**Requested Changes:**

- Adjust the claim about the model learning "architecture-level signatures" and similar should be removed or reduced (see claim evidence section). From the experiments, I agree that the model is more robust to changes different training and finetuning, but I don't think there was convincing evidence that this had to do with the architecture or that the model was necessarily extracting "signatures". The cause of the robustness is not clearly given even though the evidence of robustness itself is shown.

- The addition of autoregressive or flow-based models would help diversify. Overall, the focus on GAN-based models is concerning as these are not common anymore.

- Removal or modification of claims of what the component does in the methodology section unless you explicitly cite references that support that. You can give intuitions for the design decisions but be careful not to claim what the part actually does semantically unless you have explicit and convincing evidence for that.

- Reorganization of the experiments section for better clarity and having the comparisons first before analyzing the given method.

---

> ### Author Response · Authors · 2026-06-04
> **Response to Weakness and Suggested Changes**
>
> We sincerely thank the reviewer for the detailed evaluation and constructive feedback. We have carefully revised the manuscript to address all concerns raised in the review. For ease of assessment, all major modifications introduced in the revised manuscript are highlighted in $\textcolor{blue}{blue}$.
>
> $\textbf{W1: Limited diversity of generative models and GAN-centric evaluation}$:
> We appreciate the reviewer's concern regarding the original emphasis on GAN-based models. To address this limitation, we substantially expanded both the training and evaluation datasets to include modern generative paradigms beyond GANs. Specifically, we incorporated Qwen-Image, Stable Diffusion 3, and LlamaGen into the training set. Furthermore, we added a comprehensive evaluation on a diverse collection of unseen state-of-the-art models. These additions significantly broaden the coverage of contemporary generative paradigms and strengthen the practical relevance of GTA-Net.
>
> $\textbf{W2: Methodological claims and interpretation of learned representations}$:
> We agree that several statements in the original manuscript overstated what GTA-Net learns internally. Accordingly, we systematically revised the manuscript to remove or soften claims related to `architecture-level signatures'', `architecture-specific fingerprints'', and similar assertions. Throughout the revised paper, we now describe the learned representations in terms of their observed robustness and attribution performance rather than claiming a specific underlying mechanism.
>
> $\textbf{W3: Experimental organization and presentation}$:
> Following the reviewer's suggestion, we reorganized the experimental section to improve readability and emphasize the most important results. Comparisons against existing attribution methods are now presented more prominently, while explanatory analyses and supporting experiments have been streamlined.
>
> $\textbf{Q1: Why train our own models instead of using pretrained open-source models?}$:
> Our objective was to systematically evaluate robustness under controlled variations such as changes in initialization seeds, datasets, training duration, fine-tuning, and architectural modifications. Such controlled experiments require access to multiple model instances trained under different conditions, which is generally not possible with publicly available pretrained checkpoints. Training our own models allows us to generate consistent benchmark datasets while ensuring precise control over the factors being studied.
>
> $\textbf{Q2: Multi-Head Attention Loss and the possibility of identity mapping}$:
> We thank the reviewer for raising this important question. We clarified the role of the Multi-Head Attention loss in the revised manuscript. The purpose of $L_{MHA}$ is not to force an identity mapping but rather to regularize the attention module so that attribution-relevant information is preserved while the network is jointly optimized with supervised contrastive and classification objectives. An identity mapping would not minimize the overall optimization objective unless it simultaneously achieved optimal feature separability and attribution performance. We have added this clarification directly in the loss function section of the manuscript.
>
> $\textbf{R1: Reduce unsupported claims regarding architecture-level signatures}$:
> We carefully revised the manuscript to address this concern. Statements implying that GTA-Net explicitly learns architecture-level signatures, training-pipeline fingerprints, or objective-level fingerprints have been removed, softened, or reformulated. The revised manuscript now focuses on empirical observations showing robustness across retraining, fine-tuning, dataset shifts, and unseen models, without asserting a specific causal explanation for this robustness.
>
> $\textbf{R2: Include modern autoregressive and flow-based models}$:
> As requested, we expanded the study to include modern generative paradigms. The revised manuscript now includes autoregressive, flow-matching, multimodal, transformer-based, diffusion-based, and closed-source generative models.
>
> $\textbf{R3: Revise methodology claims and provide stronger justification for design choices}$:
> We substantially revised the methodology section to avoid unsupported semantic interpretations of individual components. In particular, we clarified the motivation behind the denoising autoencoder, residual representation, and attention mechanisms. We additionally incorporated relevant literature that motivates the use of residual representations.
>
> $\textbf{R4: Reorganize experiments and improve clarity}$:
> The experimental section has been reorganized and expanded. We improved the presentation of comparison results, clarified evaluation protocols, added modern-model experiments, and included additional discussions regarding feature separability, attribution cues, and robustness analyses.
>
> We also added a dedicated Broader Impact and Ethical Considerations section.

---

### Review · Reviewer_YwFe · 2026-05-09

**Summary Of Contributions:**

This paper proposes GTA-Net, a generative model attribution method for identifying the source model of AI-generated images. The proposed solution is reasonable: the method combines residual extraction via a denoising autoencoder, frequency-domain features using wavelet transforms, attention-based refinement, and supervised contrastive learning to learn model-level fingerprints. The paper also evaluates the method under practical variations such as cross-seed generation, cross-dataset training, fine-tuning, conditional generation, and unseen models.

**Audience:**

Yes

**Audience Explanation:**

The task is timely and important, especially because standard deepfake detection only predicts whether an image is fake, while attribution provides more useful forensic information. And high score from attribution results does tell something about the generative model behavior.

**Claims And Evidence:**

Yes

**Claims Explanation:**

The main strength of the paper is its focus on robust attribution beyond closed-set model instances, which is a meaningful step beyond prior methods. The experimental results are strong and suggest that GTA-Net captures relatively stable architecture- and training-level traces rather than simply memorizing specific generator instances.

Overall, this is an interesting and practically relevant paper with a well-motivated approach and promising results. I would lean toward a positive evaluation.

**Requested Changes:**

I would like to accept this paper as it is.

---

> ### Author Response · Authors · 2026-06-04
> **Thanking the reviewer.**
>
> We sincerely thank the reviewer for the careful reading of our manuscript and for the positive assessment of our work. We greatly appreciate the recognition of the importance of robust generative model attribution and the acknowledgement that GTA-Net represents a meaningful step beyond conventional closed-set attribution approaches.
>
> We are encouraged that the reviewer found the proposed methodology well-motivated, the experimental evaluation comprehensive, and the results convincing. We particularly appreciate the recognition of our efforts to evaluate robustness under practical scenarios including cross-seed generation, cross-dataset training, fine-tuning, conditional generation, and attribution of previously unseen models.
>
> Following the feedback received from other reviewers, we have further strengthened the manuscript through additional experiments, expanded evaluations on modern generative models, revised discussions, and clarifications throughout the paper. All major revisions are highlighted in $\textcolor{blue}{blue}$ in the updated manuscript.
>
> We are grateful for the reviewer's encouraging comments and support for publication.

---

### Review · Reviewer_K9z8 · 2026-05-21

**Summary Of Contributions:**

This paper addresses the problem of attributing an image to its source generative model, i.e., given an image, it aims to classify which generative model could have generated the image.

The proposed method (GTA-Net) has three components: (i) an abtractor network (extracts residuals from a denoising autoencoder and frequency-based features of the image based on Wavelet representations), (ii) feature extractor (lets the features interact with each other and outputs two embeddings: one for supervised contrastive learning and other for classification, (iii) multi-class classifier.

The method is trained on images generated from 11 models (8 GANs, 2 Diffusion, 1 GANformer). The denoising autoencoder is trained on real images. GTA-Net successfully attributes generated images to their corresponding generative models, even when the
images originate from models that were unseen during training.

**Strengths**

1. The attribution problem is important especially in 2026 with the rapid rise of generative image/video models.
2. The paper is well-written with all the details clearly stated.

**Weaknesses**

I found several weaknesses in the current form of this work.

1. **Use of mostly older generative models.** The authors use 8 GAN based models, 2 diffusion based models. GANs are clearly old and far from state of the art. The 2 Diffusion variants are the earliest variants and there have been massive improvements across flow-based, score-based or SDE-based models but none of them are considered. Furthermore, no evaluation of the method is conducted on any closed-source models (e.g., Nano Banana) that actually produce images indistinguishable from reality. Even the generalization eval dataset (ForgeryNet) is from 5 years ago. Hence, all of this severely limits the relevance and applicability of the proposed method.

2. **Questionable and unjustified design choices.**
    -  Choice/intuition of using denoising autoencoder is not stated clearly. Why does denoising autoencoder's residual provide any useful signal for this task? The training objective is based on MAE unlike usual MSE loss (theoretically justified and stable) - no explanation is provided.
    - How are training and validation splits created, especially for synthetic images?

3. **Unconvincing evaluation.** The model achieves ~99% accuracy on both train and validation sets. This is deeply suspicious and reeks of overfitting - there is no analysis of potential confounders, or explainability analysis of what the model's decision is based on.

4. Some recent prior work that uses Diffusion features directly to estimate attribution is missing.
[1] Zhu et al. MAID — Model Attribution via Inverse Diffusion (ICASSP 2025)
[2] Bonechi et al. Who Made This? Fake Detection and Source Attribution with Diffusion Features. Arxiv 2025.

**Audience:**

Yes

**Audience Explanation:**

Deepfake attribution continues to be an important problem, even more so now that models like Nano Banana / Flux produce such realistic images. Thus, this is a relevant problem for at least a subset of the TMLR community.

**Claims And Evidence:**

No

**Claims Explanation:**

1. The model achieves ~99% accuracy on both train and validation sets. This is deeply suspicious and reeks of overfitting - there is no analysis of potential confounders, or explainability analysis of what the model's decision is based on.

2. The evaluation is outdated in my view. The authors use 8 GAN based models, 2 diffusion based models. GANs are clearly old and far from state of the art. The 2 Diffusion variants are the earliest variants and there have been massive improvements across flow-based. It is increasingly hard to discriminate between real and synthetic images from modern models - but the evaluation/method does not consider that at all.

**Requested Changes:**

1. How does the proposed method do on some of the modern generative models? For training, it would be worth including at least two recent models (e.g., FLUX.1, SD3.5, Qwen-Image) and investigate if the proposed method works on them. For evaluation of generalisability, a strong signal could be to detect if an image comes from a strong closed model (e.g., Nano Banana) or at least of its embeddings of images from such models lie away from those from other models.

2. Interpretation: Is there a pattern in real/generated images that the model relies on? It would be good to have some visualisation and / or a quantitative experiment that there are no confounders, especially given the close to 100% accuracy on validation set.

3. Justification for architectural choices like the denoising autoencoder and why its residual is a useful feature for attribution. If this choice is based on prior work, please cite it.

---

> ### Author Response · Authors · 2026-06-04
> **Response to Weakness and Suggested Changes**
>
> We sincerely thank the reviewer for the thorough evaluation and constructive feedback. We have carefully revised the manuscript and highlighted all major modifications in $\textcolor{blue}{blue}$ throughout the paper for ease of review.
>
> $\textbf{W1: Use of mostly older generative models}$:
> We agree that evaluating only GAN-centric architectures would limit the practical relevance of the study. To address this concern, we significantly expanded the training and evaluation set to include modern generative paradigms beyond GANs. Specifically, we incorporated Qwen-Image (multimodal diffusion), Stable Diffusion 3 (flow-matching) and LlamaGen (autoregressive) into our experimental evaluation. We also added a comprehensive evaluation on unseen models in the wild, including flow-matching, autoregressive, diffusion, multimodal, and closed-source models, demonstrating the generalization capability of GTA-Net beyond the original training distribution.
>
> $\textbf{W2: Questionable and insufficiently justified design choices}$:
> We expanded the methodology section to provide a clearer motivation for the denoising autoencoder and residual representation. In particular, we now cite prior forensic attribution works showing that residual  representations often contain attribution-relevant artifacts that are less dependent on semantic image content. We additionally clarified the rationale for using MAE instead of MSE and provided details regarding the creation of training, validation, and testing splits, including the protocol used for cross-seed and cross-dataset evaluation. All corresponding modifications have been incorporated in the revised methodology section.
>
> $\textbf{W3: Potential overfitting and lack of interpretability analysis}$:
> To investigate whether GTA-Net relies on confounding semantic information, we added a dedicated discussion section to discuss potential confounders and attribution cues. We further provided TSNE plot to show how the residual-frequency representation suppresses semantic content while emphasizing attribution-relevant information. The discussion has been expanded to clarify the limitations of interpretation and to avoid unsupported claims regarding the exact nature of the learned representations.
>
> $\textbf{W4: Missing related work}$:
> We thank the reviewer for identifying these omissions. We have updated the related work section to include and discuss recent attribution methods based on diffusion representations, including \textit{MAID} (Zhu et al., ICASSP 2025) and \textit{FRIDA} (Bonechi et al., 2025).
>
>
> $\textbf{R1: Evaluation on modern generative models}$:
> Following the reviewer's suggestion, we expanded both training and evaluation to include recent generative models and paradigms. We additionally introduced an extensive evaluation on unseen real-world models, including FLUX.1, Stable Diffusion 3.5, Nano Banana, Imagen, PixArt-$\alpha$, Lumina-T2X, SANA, and several other contemporary architectures. These results demonstrate that GTA-Net remains effective even when encountering models that were not observed during training.
>
> $\textbf{R2: Interpretation and analysis of attribution cues}$:
> We appreciate this concern. To better understand the cues learned by GTA-Net, we added a new subsection in discussion. First, we highlight the t-SNE visualization of the learned embeddings, which shows compact clustering according to generator identity rather than image appearance. Second, we emphasize the cross-dataset evaluation, where GTA-Net maintains high attribution accuracy even when the target generative models are trained on a different dataset, indicating that the learned representations are not tied to a specific content distribution. Finally, we performed an additional prompt-controlled attribution experiment using 50 highly constrained facial prompts and four prompt-conditioned generators (Qwen-Image, Stable Diffusion 3, LDM, and LlamaGen). Despite generating semantically similar frontal portrait images from identical prompts, GTA-Net achieved an average attribution accuracy of 98.0%. These results suggest that GTA-Net primarily relies on generator-specific traces encoded in residual and frequency-domain representations rather than semantic image content or dataset-specific artifacts.
>
>
> $\textbf{R3: Justification of the denoising autoencoder}$:
> We have substantially expanded the discussion of the denoising autoencoder and its residual representation. The revised manuscript now cites prior forensic attribution literature demonstrating the utility of residual and frequency-domain representations for source attribution. We also clarify that the denoising autoencoder is used to obtain a content-suppressed representation that may contain attribution-relevant artifacts, rather than claiming that it explicitly extracts architectural fingerprints. This modification was made throughout the manuscript to ensure that all claims are aligned with the empirical evidence presented.

---

### Decision · Action_Editor_Bcc3 · 2026-07-04

**Recommendation:** Reject

**Additional Comments:**

While revisions introduced during the discussion phase are substantial, reviewers stated that they have fundamentally altered the manuscript's scope, claims, and organization. Reviewer WKZ3 and Reviewer K9z8 both pointed out that the addition of flow-matching, autoregressive models, and the systematic rewriting of core claims warrant a thorough, full round of re-review. Moreover, concerns about potential overfitting, given the exceptionally high classification accuracy (~99%), require a more rigorous, fresh evaluation cycle.

Therefore, the submission is rejected. However, because the direction of the revision is highly promising and the task formulation holds clear practical relevance, the authors are strongly encouraged to resubmit this work as a fresh submission.

**Audience:**

Yes

**Audience Explanation:**

The problem is a highly timely and crucial problem within the forensics community, especially given the rapid proliferation of realistic generative models. Understanding how statistical artifacts is important to trace the source of generated imagery.

**Claims And Evidence:**

No

**Claims Explanation:**

The authors have put significant effort into addressing the reviewers' initial concerns. In the revised manuscript, they expanded the experimental evaluation to include modern generative paradigms (such as FLUX.1, Stable Diffusion 3.5, and Qwen-Image) and tempered their overclaiming regarding the explicit learning of "architecture-level signatures."